# Learning Control Admissibility Models with Graph Neural Networks for Multi-Agent Navigation

**Chenning Yu**
UCSD
chy010@ucsd.edu

**Hongzhan Yu**
UCSD
hoy021@ucsd.edu

**Sicun Gao**
UCSD
sicung@ucsd.edu

**Abstract:** Deep reinforcement learning in continuous domains focuses on learning control policies that map states to distributions over actions that ideally concentrate on the optimal choices in each step. In multi-agent navigation problems, the optimal actions depend heavily on the agents' density. Their interaction patterns grow exponentially with respect to such density, making it hard for learning-based methods to generalize. We propose to switch the learning objectives from predicting the optimal actions to predicting sets of admissible actions, which we call control admissibility models (CAMs), such that they can be easily composed and used for online inference for an arbitrary number of agents. We design CAMs using graph neural networks and develop training methods that optimize the CAMs in the standard model-free setting, with the additional benefit of eliminating the need for reward engineering typically required to balance collision avoidance and goal-reaching requirements. We evaluate the proposed approach in multi-agent navigation environments. We show that the CAM models can be trained in environments with only a few agents and be easily composed for deployment in dense environments with hundreds of agents, achieving better performance than state-of-the-art methods.

## 1 Introduction

Multi-agent navigation is a longstanding problem with a wide range of practical applications such as in manufacturing [1, 2], transportation [3], and surveillance [4]. The goal is to control many agents to all achieve their goals while avoiding collisions and deadlocks. Such problems are multi-objective in nature, and the control methods depend heavily on the density of the agents, with the computational complexity growing exponentially in such density [5, 6, 7, 8].

Recently, reinforcement learning approaches have shown promising results on multi-agent navigation problems [9, 10, 11, 12, 13], but there are still two well-known sources of the difficulty. First, balancing safety and goal-reaching requirements often lead to ad hoc reward engineering that is highly dependent on the environments and the density of the agents [14, 15]. Second, it is shown that such neural control policies leaned on RL approaches are hard to generalize when the agent density in the environment changes, which leads to distribution drifts. The core difficulty for generalizability is that the RL approaches are designed to capture the optimal actions specific to the training environments and rewards. Nevertheless, such optimal actions may change rapidly when the density and interaction patterns change during deployment. The unnecessary optimization of control policies in the training environments makes it fundamentally hard to transfer the learned results to new environments and achieve compositionality in the deployment to varying numbers of agents.

We propose a new learning-based approach to multi-agent navigation that shifts the focus from learning the optimal control policy to a set-theoretic representation of *admissible* control policies to achieve compositional inference. By decoupling the multi-agent navigation tasks, we avoid the ad-hoc reward engineering using a simple goal-reaching preference function and a learnable set-theoretic component for collision avoidance. We use graph neural networks (GNNs) to represent the set of admissible control actions at each state as Control Admissibility Models (CAM), which are trained with a small number of agents during training with sparse rewards. In online inference, we compose the CAMs and apply goal-reaching preference functions to infer the specific action to take for each agent. We show that CAMs can be learned purely from the data collected online in the standard RL

6th Conference on Robot Learning (CoRL 2022), Auckland, New Zealand.

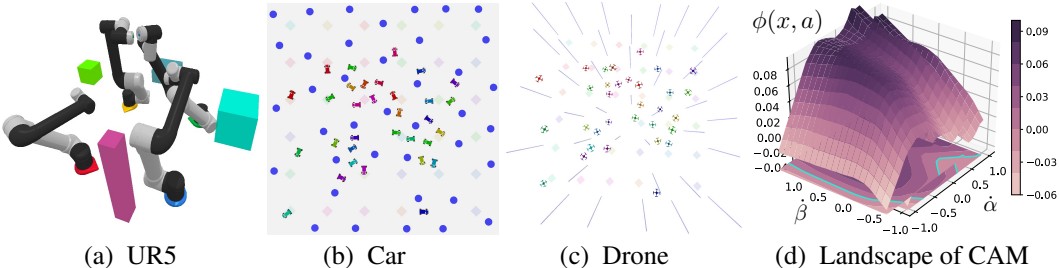

| (a) UR5 | (b) Car | (c) Drone | (d) Landscape of CAM |
|---|---|---|---|

Figure 1: **(a-c)** Illustrations of the multi-agent environments. We show videos in the supplemental materials. **(d)** An example landscape of the proposed CAM $\phi$ in the Drone environment. The $\dot{\alpha}$-axis and $\dot{\beta}$-axis, are two dimensions in the action space, corresponding to the rotation rates in pitch and roll. We project the landscape into a 2D plane at the bottom. The blue line on the plane denotes the zero level set, which divides the action space into the admissible and inadmissible sets.

setting, with no expert or human guidance. We propose relabelling through backpropagation (Section 2.3) for the CAMs to learn rich information from the transitions even given sparse rewards. As shown in Figure 1(d), the learned models of the admissible actions are typically complex and suitable for GNN representations. Moreover, in online inference, we decompose the state graph of the agent into much smaller subgraphs and aggregate the CAMs from them. Such compositionality allows us to train CAMs with only a small number of agents, directly deploy them in much denser environments, and achieve generalizable learning and inference.

We evaluate the proposed methods in various challenging multi-agent environments, including robot arms, cars, and quadcopters. Experiments show that the CAM generalizes very well. While the training environments only have at most 3 agents, the learned CAM models can be directly deployed to hundreds of agents in comparable environments and achieve a very high success rate and low collision rate. We analyze the benefits of the proposed methods compared to state-of-the-art approaches. Furthermore, we show a zero-shot transfer to a multi-agent chasing game, demonstrating that the trained CAM generalizes to other tasks that are not limited to navigation.

***Related Work.*** **Multi-Agent Navigation.** Classical Multi-Agent Path Finding is one of the most popular definitions of the multi-agent navigation task, where the state space lies in a pre-constructed graph shared among agents [16]. Under this setting, heuristic-based methods have been used to generate high-quality solutions, such as Priority-Based Search and Conflict-Based Search [17, 18]. Learning-based methods have also been proposed to accelerate the planning time, using CNN and GNN [19, 20, 21]. In this work, we focus more on the continuous state space with reactive planning. Traditional methods, including Optimal Reciprocal Collision Avoidance [22], Buffered Voronoi Cells [23], and Artificial Potential Functions [24, 25, 26, 27], are designed manually to address specific structural properties. On the other hand, learning-based multi-agent methods using reinforcement learning and imitation learning [9, 10, 11, 12, 13, 28, 29], may or may not generalize when there is covariant shift for test tasks. Recently, MACBF [30] incorporates multi-agent safe certificates [31, 32, 33] into a learning-based framework, which shows the generalization capability while preserving the safety properties. Nevertheless, in [30] the learning of the neural controller requires a reference controller, which may constrain the potential of the neural controller, when the actions dissimilar to the reference actions could yield better rewards and safer results. Though learned in an online fashion, our work only assumes sparse reward and avoids the reward hacking problem.

**Graph Neural Networks.** Graph Neural Networks (GNN), including Graph Convolutional Networks [34], Message Passing Neural Networks [35], Interaction Networks [36], are deep neural networks that operate on graph-structured data [37]. Graph neural networks are permutation-equivariant to the orders of nodes on graph [38, 39], which is important for homogeneous multi-agent problems since the order among agents should not matter. In robotics, the effectiveness of GNN has been shown on tasks such as path planning [19, 20], motion planning [40, 41], coverage and exploration [42, 43], and SLAM [44]. In this work, we use GNN as an architecture to implement CAM due to the multi-agent navigation problem can be modeled as a graph-based problem in nature. However, we believe that the proposed CAM could generalize to other inputs that are not graphs.

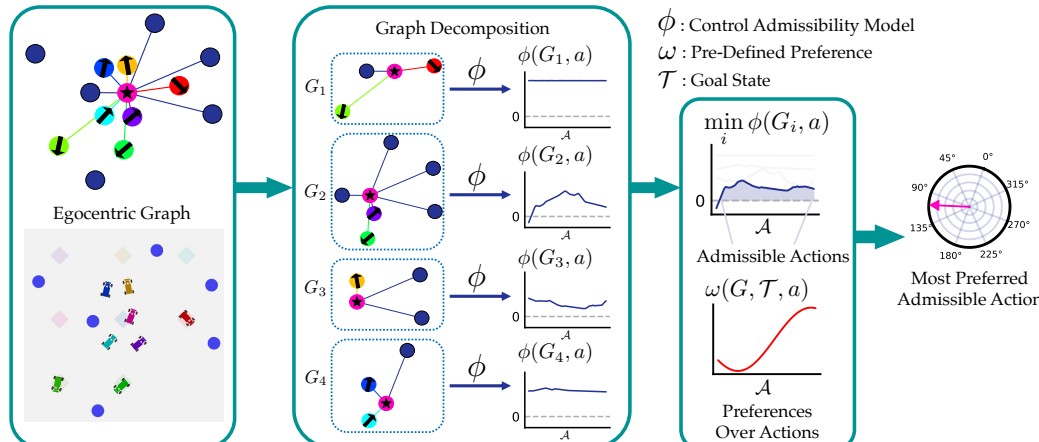

Figure 2: An overview of the proposed CAM approach. At each time step, the state of each agent is an egocentric graph. The graph is further decomposed at inference time into a set of subgraphs to follow the training data distribution. The CAM takes these subgraphs and outputs the admissibility scores along with a set of sampled candidate actions. We filter out the inadmissible actions by checking whether the minimum of the scores is below 0. Given a predefined preference over actions, the model outputs the most preferred action in the admissible set for each agent.

## 2 Control Admissibility Models

We present Control Admissibility Models (CAMs), a general representation of the feasible control set. CAM is task-compositional, robust to reward design, and can be trained with an online pipeline similar to model-free RL. With graph decomposition, CAM can plan a high-quality solution for a large number of robots, which could be significantly out of the distribution of the training tasks.

### 2.1 Control Admissibility Models (CAMs)

We define CAMs as a general representation of the feasible actions. In general, CAMs are function approximators that behave as the characteristic functions of such sets. By evaluating the CAM values we can efficiently determine whether an action is feasible. Formally,

**Definition 1** (Control Admissibility Models and Admissible Set). *Given CAM $\phi$, state $x$, action space $\mathcal{A}$, the admissible set is defined as $\Delta(\phi, x) : \{a \mid a \in \mathcal{A}, \phi(x, a) \geq 0\}$.*

**Compositionality of CAMs.** An important feature of CAMs is that they can be composed together. Given two CAMs $\phi_1$ and $\phi_2$, we want to find the admissible actions that satisfy both these two CAMs:

$$\{a \mid \phi_1(x, a) \geq 0\} \cap \{a \mid \phi_2(x, a) \geq 0\}$$

**Proposition 1** (Composition). *If $\phi_1$ and $\phi_2$ are CAMs, then $\Delta(\phi_3, x)$ defines $\Delta(\phi_1, x) \cap \Delta(\phi_2, x)$, given $\phi_3 = \min\{\phi_1, \phi_2\}$.*

### 2.2 Graph Neural Networks for CAMs

Though CAM can be applied to various single-agent problems (see Section 3.1 as an example), we focus on multi-agent problems in this paper. At each timestep, the state for each agent is an egocentric graph $G = \langle V, E \rangle$, where $V$ are the vertices and $E$ are the edges. Vertices $V$ is composed of agent vertices $V_a$ and static obstacle vertices $V_o$. Edges $E$ are connecting all the neighbor vertices to the current agent, i.e. $E = \{(v_a^j, v_a^i) | a_j \in \mathcal{E}(a_i) \cup \{(v_o^j, v_a^i) | o_j \in \mathcal{E}(a_i)\}$, where $\mathcal{E}$ denotes the neighbor vertices. The one-hot vector is used as the features of $V$, denoting the types of vertices. For the features of edges $E$, we use relative positions and the states of the two connected vertices, along with a one-hot vector to indicate the edge types. Zero paddings are adopted if necessary [45]. We provide more details on the representation of graphs in the Appendix.

The CAM $\phi$ has two components: The first component is a GNN layer, which transforms the graph $G$ to a hidden state vector $h$ for each agent. The second component is a fully-connected layer $\hat{f}$, which

takes the hidden state $h$ and an action $a$, and predicts the admissibility score $\phi(G, a) = f(h, a)$. Such a design enables fast computation of the admissibility scores for all the possible state-action pairs. We can merge the computation of the hidden state for all state-action pairs which share the same input graph into one forward passing in the GNN layer.

We describe the GNN layer with the following details. The vertices and the edges are first embedded into latent space as $m^{(0)} = g_m(v), n^{(0)} = g_n(e)$, given fully-connected layers $g_m, g_n$. Taking $m, n$, the GNN aggregates the local information for each vertex from the neighbors through $K$ GNN layers. For the $k$-th layer, it performs the message passing with 2 fully-connected layers $f_m^{(k)}$ and $f_n^{(k)}$:

$$
\begin{aligned}
m_i^{(k+1)} &= m_i^{(k)} + \max\{f_m^{(k)}(n_l^{(k)}) \mid e_l : (v_j, v_i) \in E\}, \forall v_i \in V \\
n_l^{(k+1)} &= n_l^{(k)} + f_n^{(k)}(m_i^{(k+1)}, n_l^{(k)}), \forall e_l : (v_j, v_i) \in E
\end{aligned}
\tag{1}
$$

We assign the hidden state $h$ with the final node embedding $m_i^{(K)}$, where $i$ corresponds to the current agent. We use max as the aggregation operator to gather the local geometric information due to its empirical robustness to achieve the order invariance [30, 39]. We use the residual connection to update the vertex embedding and edge embedding due to its simplicity and robust performance for deep layers [46, 47]. Moreover, each agent's hidden state $h$ is only updated based on its egocentric graph, which is entirely invariant to the other agents' hidden states. This architecture decentralizes the decision-making of the whole swarm.

## 2.3 Training CAMs in Reinforcement Learning

We train CAMs in the same online setting as model-free reinforcement learning procedures. At each time step, the agent perceives the observation from the environment and takes action. The state-action pairs for every transition are labeled as safe or unsafe and then appended to the replay buffer. The CAM is updated every time a certain number of transitions are collected. The main differences between our training approach and the standard RL methods are three-fold: **(i)** instead of using an actor-network, the agent chooses the action among the sampled actions based on the admissibility score generated by CAM. **(ii)** the label is binary for each state-action pair instead of calculating the cumulative future rewards. **(iii)** the training objective is non-bootstrapped, which avoids the overestimation issue and is more stable. We design these three improvements around the underlying structure of the admissibility problem. We provide further explanations as follows.

---
**Algorithm 1** CAM Algorithm for Training
---
1: **Input**: Preference function $\omega$, exploration noise $\mathcal{N}$
2: Initialize CAM $\phi$
3: Initialize replay buffer $R$
4: **for** episode = 1, M **do**
5:     **for** t = 1, T **do**
6:         Sample candidate actions $\{a\} \subseteq \mathcal{A}$
7:         $\{G\}, \{\mathcal{T}\} \leftarrow$ all agents' current states and goal states
8:         Select action for each agent according to $\phi(G, a), \omega(G, \mathcal{T}, a)$, and $\mathcal{N}$
9:         Execute the selected actions and observe the next states $s_{t+1}$
10:         Label the transition $(s_t, a_t, s_{t+1})$ with $y_t$ by checking whether $s_{t+1}$ is in danger set
11:         Store transition $(s_t, a_t, y_t, s_{t+1})$ in $R$
12:     Relabel $R$ through backpropagation
13:     Update $\phi$ using Equation 2
---

**Sparse Reward Setting.** We model the problem as the standard Markov Decision Process (MDP) $\mathcal{M} = (\mathcal{S}, \mathcal{A}, T, r, \gamma)$ [48]. $\mathcal{S}$ is the state space represented as graphs $G$. The action space $\mathcal{A}$ is continuous. The reward $r$ is sparse: it only returns non-zero values when the agent enters the region to avoid and the region to reach. These certain regions can also be dynamic, i.e., danger regions centering around other agents.

**Preference Function.** Given current state $G$ and goal state $\mathcal{T}$, we assume a preference function over actions, i.e., $\omega(G, \mathcal{T}, \cdot)$, is given. The preference function solely focuses on the goal-reaching. For instance, it can simply be the L2 distance from the next state to the goal, since CAM will achieve the obstacle avoidance. In the experiments, we try to choose the preference functions as simple as possible. We refer readers to the Appendix for more details.

**Action Selection via Sampling.** At every timestep, the agent chooses an action among a batch of actions sampled from the action space. Given a state $x$, the admissibility value every action $a$ is computed with $\phi(x, a)$. If there exist one or multiple actions that satisfy $\{a \mid \phi(x, a) \geq 0\}$, given the goal state $\mathcal{T}$ and a preference $\omega(x, \mathcal{T}, \cdot)$ over actions, we choose the action that has the highest preference score in the admissible set. Namely, $\arg\max_a \{\omega(x, \mathcal{T}, a) \mid \phi(x, a) \geq 0\}$. If there does not exist any admissible action, in order to fulfill the admissibility property maximally, the agent chooses the action that has the highest admissibility score, $\arg\max_a \{\phi(x, a)\} + \mathcal{N}$, where $\mathcal{N}$ is an exploration noise. We model the exploration noise under the uniform distribution in our experiment.

**Training Loss of CAM.** Given admissible transitions $R_0 \subseteq \mathcal{S} \times \mathcal{A}$ and inadmissible transitions $R_d \subseteq \mathcal{S} \times \mathcal{A}$, we train the CAM with the following objective:

$$
L_B = \frac{1}{|R_0|} \sum_{(x,a) \in R_0} \text{ReLU}(\gamma_1 - \phi(x, a)) + \frac{1}{|R_d|} \sum_{(x,a) \in R_d} \text{ReLU}(\gamma_2 + \phi(x, a))
$$
$$
+ \frac{1}{|R_0|} \sum_{(x,a) \in R_0} \text{ReLU}(\gamma_3 - \dot{\phi}(x, a) - \lambda\phi(x, a)) \quad (2)
$$

where $\gamma_1, \gamma_2, \gamma_3 \geq 0$ are the coefficients to enlarge the margin of the learned boundary. Instead of simply classifying the transitions with the first and the second loss terms, we encourage the forward invariance property by adding the third term, similar to Control Lyapunov Functions and Control Barrier Functions [31]. In practice, we approximate $\dot{\phi}(x, a)$ by $\phi(x', a') - \phi(x, a)$, where $(x', a')$ is the next state-action pair on the trajectory. Intuitively, once all pairs of consecutive transitions satisfy this condition, the transitions would form a forward invariant set, and any trajectory starting from inside the invariant set will never cross the admissible boundary. The $\lambda\phi(x, a)$ term encourages the trajectory to be asymptotically admissible, even if the agent enters the inadmissible region sometimes.

**Relabeling Through Backpropagation.** We first label the collected transitions as admissible and inadmissible based on whether the next state is in the danger region. Once the whole trajectory terminates, we relabel these transitions through backpropagation. Intuitively, suppose the agent encounters the collision along the trajectory. In that case, it means that starting from some timestep, it enters the inadmissible region and fails to get back to the admissible set (similar to Intermediate Value Theorem [49]). Without relabeling, the labels of the transitions from this step to the collision event will be admissible, which is incorrect. To solve this issue, we relabel these critical transitions.

A state-action pair $(x, a)$ on the trajectory is relabelled as inadmissible, if **(i)** the next state-action pair $(x', a')$ is inadmissible, and **(ii)** no admissible action is found at state $x'$ - namely, $\max_{a'} \phi(x', a') < 0$. Intuitively, the relabelling propagates the inadmissibility from where the agent enters the danger region back to a state, where the agent can make a decision leading to admissible regions. The first condition ensures the optimistic behavior of the CAM. We only relabel those transitions which stay in the inadmissible region and encounter danger in the future. Such optimism is important when no feasible trajectory exists at the early training stage. It encourages the agent to explore instead of predicting almost every transition to be inadmissible. The second condition ensures that there exist no admissible actions approximately by inspecting the admissibility score for all the actions. Furthermore, for the transitions that have potential admissible actions, they will not be relabelled as inadmissible easily at the early training stage since $\max_{a'} \phi(x', a') < 0$ is hard to meet for a moderately initialized CAM.

## 2.4 Online Inference with Graph Decomposition

At inference time, the agent behaves in the same manner as the training stage, except that there is no exploration noise during inference. When the CAM is applied directly to a large swarm of agents, however, we find that the performance degrades significantly due to the distribution shift. Such degradation occurs considerably when the inference task has a much higher density of agents. To solve this issue, we leverage the compositional property of CAMs and propose graph decomposition.

**Decomposition on Large Graphs.** We can break down the whole graph $G$ into subgraphs $\{G_k\}$, where these subgraphs satisfy the training distribution, e.g., the number of agent vertices does not exceed a certain value. We construct the subgraphs repeatedly until all the edges in $G$ appear at least once in the subgraphs. For each subgraph $G_k$, and action $a$, we calculate the admissibility score as $\phi(G_k, a)$. Though each subgraph defines a less strict subtask for the agent to perform, the admissible set for the original task should lie in the intersection of all these admissible sets of the subtasks. As

a result, these values on the subgraphs can be composed to represent the admissibility score on the entire graph $\phi(G, a)$, which could be totally out of distribution. To compose the admissibility values, we follow Proposition 1 and use $\phi(G, a) = \min_{G_k \subseteq G}\{\phi(G_k, a)\}$.

---

**Algorithm 2** CAM Algorithm for Inference

---

1: **Input**: CAM $\phi$, preference function $\omega$, distribution $P$ of training data
2: **for** t = 1, T **do**
3:     **for** each agent's current state $G$, goal state $\mathcal{T}$ **do**
4:         Sample candidate actions $\{a\} \subseteq \mathcal{A}$
5:         Initialize $\phi(G, a)$ as $\infty$ for all sampled candidates
6:         **repeat**
7:             Sample $G_i \subseteq G$, where $G_i \sim P$
8:             $\phi(G, a) = \min(\phi(G, a), \phi(G_i, a))$ for all candidates $a$
9:         **until** all edges on $G$ are sampled at least once
10:         Select action for the agent according to $\phi(G, a)$ and $\omega(G, \mathcal{T}, a)$
11:     Execute the selected actions and observe the next states $s_{t+1}$

---

# 3 Experiments

## 3.1 Proof of Concept: CAM for Single Agent Environment

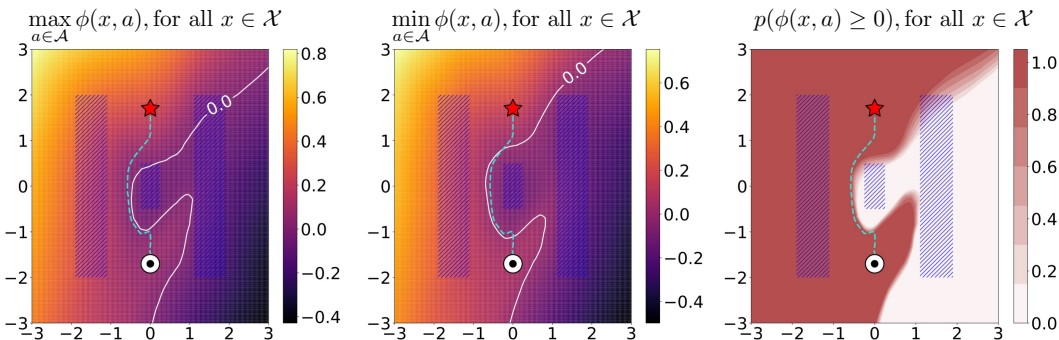

Figure 3: A proof of concept for the proposed CAM in a single-agent environment. The agent aims to reach the goal while avoiding obstacles. The black point denotes the starting state, and the red star denotes the goal state. The shaded blue regions denote the obstacles. Given the learned CAM $\phi$, action space $\mathcal{A} : \{-0.1, 0.1\}^2$, and an arbitrary state $x \in \mathcal{X} : \{-3, 3\}^2$, we illustrate $\max_{a\in\mathcal{A}} \phi(x, a)$, $\min_{a\in\mathcal{A}} \phi(x, a)$, and $p(\phi(x, a) \geq 0)$ respectively. We show that the learned CAM never leaves the admissible set, i.e. $\{x \in \mathcal{X} : \max_{a\in\mathcal{A}} \phi(x, a) \geq 0\}$, along the trajectory.

We begin by analyzing the CAM agent in a single-agent environment. We perform such analysis using a 2D minimal example environment, where we place three danger regions to form narrow passages, as shown in Figure 3. We illustrate $\max_{a\in\mathcal{A}} \phi(x, a)$, $\min_{a\in\mathcal{A}} \phi(x, a)$, and $p(\phi(x, a) \geq 0)$ respectively. Intuitively, $\max_{a\in\mathcal{A}} \phi(x, a)$ indicates whether there exists any admissible actions that drive the agent to the admissible set. $\min_{a\in\mathcal{A}} \phi(x, a)$ indicates the possibility to enter the inadmissible sets if actions are picked randomly. As what we show in the figure, the trajectory (blue dashed line) does not intersect with the admissibility boundary, i.e. $\{x \in \mathcal{X} : \max_{a\in\mathcal{A}} \phi(x, a) = 0\}$. This example stays consistent with the forward-invariance objective for learning the CAM: if the agent executes an admissible action $a \in \mathcal{A} : \phi(x, a) \geq 0$ at every time step, then it will never leave the admissible set.

## 3.2 Multi-agent Navigation Experiments

**Experiment Setup.** We evaluate our methods under 4 types of environments: UR5, Car, Dynamic Dubins, and Drone. We briefly describe these 4 environments as follows: **(i)** The UR5 environment is implemented with PyBullet [50] and contains 4 decentralized 5D robot arms with cubic obstacles. We randomly select solely 2 robot arms from these four arms during training. We fix the configurations of the static obstacles and the goal states of the arms. In order to make the test task solvable, the initial

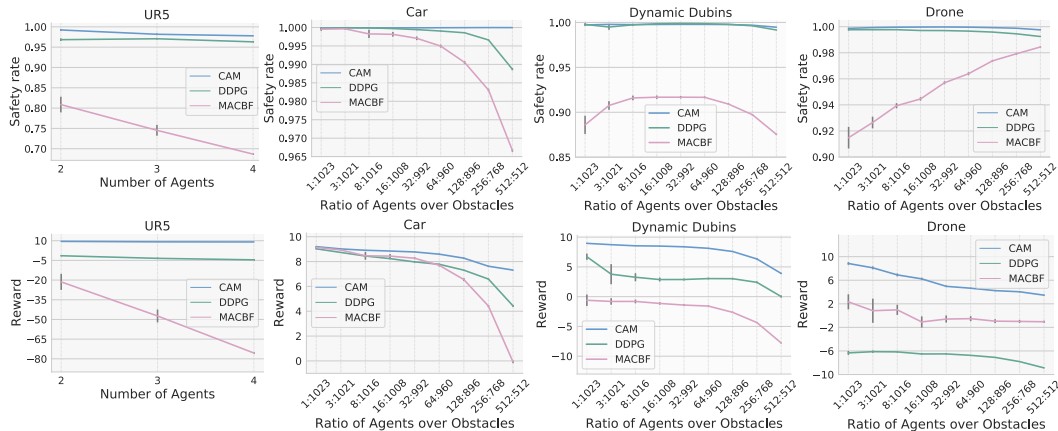

Figure 4: We show the performances of CAM and prior methods in the UR5, Car, Dynamic Dubins, and Drone environment, where each method is trained with 3 agents during training but tested up to 512 agents. CAM substantially outperforms prior approaches in all the environments.

states are randomized during training and fixed during testing. **(ii)** For the Car environment, each agent follows the 2D Dubins' car model [51], and controls a 1D action. The static obstacles are circles with a fixed radius. **(iii)** The Dynamic Dubins agent controls a 2D action and follows a modified Dubins' Car dynamic which enables the control of acceleration. **(iv)** For the Drone environment, each agent follows a 3D quadrotor model adopted from [52] and controls a 4D action. Each obstacle is a cylinder with infinite height. For both Car and Drone environments, we only train on environments with 3 agents, randomized initial states and goal states, and a set of random obstacles. The map size is fixed as 3x3 during training while varying during testing. We provide full details of system dynamics and environments in the Appendix.

To evaluate our CAM approach thoroughly, we design diverse scenarios for all four environments. For the UR5 environment, we make the number of agents vary from 2 to 4. For scenarios with 2 or 3 arms, the arms are selected randomly from the four decentralized arms. For the other environments, we fix the map size as 32x32 and vary the ratio of agents over obstacles from 1:1023 to 512:512. For each setting, we average the performance over 100 randomly generated test cases.

**Baselines.** The baseline approaches we compare with include DDPG [53] and MACBF [30]. DDPG is a standard RL approach to learning safe behavior through maximizing rewards. MACBF is a state-of-the-art approach for multi-agent safe control problems, which learns a policy jointly with neural safe certificates. We reimplement all the baselines with GNNs for a fair comparison. Note that our GNN implementation for DDPG is a multi-agent algorithm, since each agent is aware of the other agents by taking the egocentric graph as the input. We refer reader to Appendix for more details.

**Evaluation Metrics.** We evaluate our methods based on two metrics: the safe rate and the average reward. The safe rate counts the number of collisions for each agent at every time step and averages over all the agents through the whole trial [30]. At each time step, the reward is -1 if the agent collides with another agent or the static obstacle. A +10 reward will be given at the end of the trajectory, if the agent reaches its goal at any time step along the trajectory. A small negative constant of -0.1 is added to the reward to encourage shorter paths for goal-reaching. The reward is accumulated throughout the trial and averaged over all the agents.

**Overall Performance.** In Figure 4, we demonstrate the overall performances for the UR5, Car, Dynamic Dubins, and Drone environments. Our method shows the generalization across different environments and densities of agents and obstacles. Though trained with only 3 agents, the safety rate of CAM remains nearly 100% for all the environments, up to 512 agents. Even if the density of agents and obstacles deviates from the training environment, our CAM approach can avoid the distribution shift, using decomposition on large graphs and the compositionality of CAMs.

On the other side, though our implementation for MACBF and DDPG can take an arbitrary number of agents using GNNs, the performance degrades significantly when the distribution of input graphs shifts in test environments. We also inspect that the averaged reward for all the methods decreases in most cases when the number of agents increases. It is due to more agents failing to reach the goal when the timeout happens since it takes more time for agents to avoid each other. We encourage the reader to view the video demonstrations in the supplementary material for all 4 environments.

### 3.3 Zero-Shot Transfer to Chasing Game

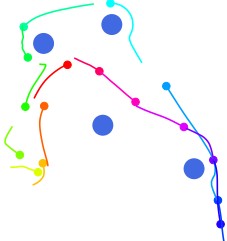

| Method | CAM | CAM w/o decomposition |
|---|---|---|
| Safety Rate (Car) | 0.99±0.00 | 0.90±0.03 |
| Reward (Car) | 3.10±1.20 | -19.98±7.08 |
| Safety Rate (Drone) | 0.989± 0.005 | 0.988±0.008 |
| Reward (Drone) | 6.84±1.12 | 6.54±2.15 |

Table 1: Left: A snapshot of a trajectory in the chasing game with 16 Drone agents from a bird's-eye view. The agents are the colorful points, and the obstacles are the blue circles of a larger size. Each agent aims to chase another agent without collision. Right: Performances of CAMs with and without graph decomposition on 64 agents, which shows the effectiveness of the graph decomposition.

In this section, we study the generalization of our CAM model for zero-shot transfer to other multi-agent tasks, e.g., the chasing game. In the chasing game, each agent chases another agent to pursue. The target agent is assigned randomly at the beginning of every episode. The agent's goal is to maintain the distance to the assigned agent, with no collision with agents and obstacles. We fix the number of agents to 64 in the experiment. This new task is substantially different from the task for training, where the assigned goal is fixed. We provide more details of the setting in the Appendix.

In Table 1, we show that our CAM method can adapt to the new task effectively due to its safety and goal-reaching decoupling property, preserving a relatively high safety rate and reward. In addition, we compare our method to the CAM without the graph decomposition. We observe that the graph decomposition works effectively in both environments. The performance of the Car environment benefits more from the graph decomposition. This advancement is because the density of Car agents in 2D is higher than that of the Drone agent in 3D. The higher density makes the test task more unlike the training task, which explains why the graph decomposition is more beneficial for Car.

## 4 Limitations and Failure Modes

While the framework is generalizable in terms of scalability, it has limitations. Since the proposed GNN architecture is entirely decentralized, it disables the agents from communicating and coordinating with each other in the swarm. As a result, it could be challenging for agents to identify and avoid critical situations requiring interactions among agents, e.g., dead-locks and states with very few feasible actions. As shown in Figure 4, there is a slight drop in the safety rate of the Drone environment when the number of agents increases to 512. To understand such a failure mode, we inspect these trajectories and find that the corresponding CAM values are all negative at several consecutive preceding states right before the collision. Such behavior indicates a very low probability of finding feasible actions in these states, which validates the necessity for inter-agent communication or global coordination when the density of agents is very high in local regions. We believe that a preference function considering the global coordination can be helpful for this issue.

## 5 Conclusion

We proposed new learning-based control methods for multi-agent navigation problems by shifting the focus from training optimal control policies to training CAMs that implicitly represent a set of feasible actions. We avoid the reward engineering by decoupling the multi-agent navigation tasks using a simple goal-reaching preference function and a learnable CAM for collision avoidance. Our methods use GNNs to represent the CAM which takes egocentric graphs as inputs and proposes the relabelling with backtracing to learn from rich information even given sparse reward. In online inference, we propose the graph decomposition to deal with the covariate shift, which achieves a notably high success rate and low collision rate. While unconventional compared to the types of models routinely used in multi-agent reinforcement learning, we demonstrate that CAM has several useful properties and is particularly effective in multi-agent settings that require compositionality and generalization for scaling at inference time. In future work, we will study how to incorporate multi-agent communication and test its effectiveness in real-world scenarios.

**Acknowledgments**

We thank the anonymous reviewers for their helpful comments in revising the paper. This material is based on work supported by the United States Air Force and DARPA under Contract No. FA8750-18-C-0092, AFOSR YIP FA9550-19-1-0041, NSF Career CCF 2047034, and Amazon Research Award.

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
