# OpenReview forum: "Learning Control Admissibility Models with Graph Neural Networks for Multi-Agent Navigation"
_robot-learning.org/CoRL/2022/Conference — CoRL 2022 Poster_

### Official Review · Reviewer_XUE1 · 2022-08-01

**Originality:** Fair
**Technical Quality:** Fair
**Clarity Of Presentation:** Good
**Impact:** 3

**Recommendation:**

Weak Reject: I recommend rejecting the paper, but will not argue for my recommendation if the majority of other reviewers have a different opinion.

**Summary:**

The authors proposed a strategy to compute admissible sets for multi-agent systems, i.e., a set of actions that agents can take without compromising safety. These sets are computed using GNN which naturally takes into account interaction between different agents. For training, each state-action pair is labeled based on the state of the system and then a relabeling state is applied to make sure that wrong actions that led to unsafe regions are not labeled as safe. Finally, the authors showed that the proposed approach trained in scenarios with only a few agents generalizes to settings where several agents are interacting.

**Issues:**

The method is well presented, but it has no theoretical guarantees (this is totally fine as computing invariant sets is very hard already for a single agent!). So I focused on the examples and I would like to see a few changes to confirm that the method is working:
1) What is the model used for the single agent environment? I would like to see an agent that has a lot of inertia, e.g., a car where you control the acceleration rate and the maximum acceleration rate is saturated. This would allow us to understand if the re-labeling strategy is working and if you are able to catch failure early on.
2) Figure 3 should have some information about the velocity. If you are on the boundary of the set with an outside pointing velocity vector the set is not invariant. It would be nice to see if the set that you are computing is actually an invariant (or a good approximation)
3) Why the set in Figure 3 is not symmetric? As we know that subsets to invariant sets are not invariant, it is unclear if the set shown in Figure 3 is forward-invariant.
4) In the Dubins car examples, are you controlling the velocity? It would be more convincing to control the acceleration and to have a saturation on the acceleration. For the dynamics, you can simply assume that the velocity is the integral of the acceleration. If you are currently controlling the velocity, could you repeat the experiments when controlling the acceleration? I expect to see more collisions.

**Quality Of The Limitations Section:**

Limitations are addressed clearly

**Reviewer Expertise:**

3: The reviewer is fairly confident that the evaluation is correct

**Robotics Focus:**

Highly relevant to robotics but no hardware experiments

**Strengths And Weaknesses:**

Strengths
1) Using a GNN for multi-agent navigation makes sense and enables scalability
2) The re-labeling strategy allows the authors to catch failure early on

Weaknesses
1) The authors are basically trying to compute invariant sets. It would be nice to introduce a formal definition, clearly state the objective, and provide an analysis (if possible).
2) From the first example, it is unclear if the method is actually able to approximate forward invariant sets, even in the very simple example shown in the experiment section.

**Summary Of Recommendation:**

I think that the paper is well-written and that the method is well-designed. The objective of this work is pretty ambitious as the authors are approximating invariant sets. Clearly, no guarantees can be provided for this problem, so the authors performed a set of experiments to empirically show that the method is working. I think that further experiments are needed. In particular, I would like to see the method applied to high inertia systems where the control action is saturated, see the Issues section for further details.

---

> ### Author Response · Authors · 2022-08-23
> **Response to Reviewer XUE1**
>
> **Comment:**
>
> Thank you for your careful reading. We appreciate your constructive suggestions. The materials mentioned below, including the updated Appendix and a new video, are attached to this response.
>
> > What is the model used for the single agent environment? I would like to see an agent that has a lot of inertia, e.g., a car where you control the acceleration rate and the maximum acceleration rate is saturated. This would allow us to understand if the relabeling strategy is working and if you are able to catch failure early on.
>
> The model in the single agent environment is the single integrator. Based on the suggestion, we have provided an additional single agent example in Appendix J. In this new example, the car controls the acceleration rate and the heading angular velocity, and the maximum acceleration rate is saturated.
>
> We illustrate the behavior of the learned CAM in Figure 9. We have also attached a video to this response. As shown, the CAM agent learns to continuously decrease the velocity and stop in advance before a potential collision happens.
>
> In addition, we also discuss how the CAM learns such behavior by inspecting how the success rates and the number of relabelled transitions change during training. We find that the success rate improvement is evidently related to the number of relabelled transitions. Please check Appendix J.2 for further details.
>
> > Figure 3 should have some information about the velocity. If you are on the boundary of the set with an outside pointing velocity vector the set is not invariant. It would be nice to see if the set that you are computing is actually an invariant (or a good approximation)
>
> Thank you for the suggestion. We have provided the velocity vectors of the boundary and danger states in Appendix I. In Figure 8, we show that the trajectory at the inference time never leaves the admissible region, since all the states along the trajectory obey the forward invariance. We find that around 9‰ of the states in the boundary violate the forward-invariance property. These states may or may not be seen during training. We do not require all the safe states in the state space to be forward invariant. Instead, the CAM only focuses on those states that might appear during inference. In this example, our goal is to ensure the states along the shown trajectory obey the forward invariance, which is accomplished successfully.
>
> > Why the set in Figure 3 is not symmetric? As we know that subsets to invariant sets are not invariant, it is unclear if the set shown in Figure 3 is forward-invariant.
>
> As shown in Algorithm 1 in the main paper, we train the CAM only for those transitions collected from past trajectories, which may or may not cover the whole joint space of states and actions. As a result, the agent may meet fewer transitions on the left side during training.
>
> > In the Dubins car examples, are you controlling the velocity? It would be more convincing to control the acceleration and to have a saturation on the acceleration. For the dynamics, you can simply assume that the velocity is the integral of the acceleration. If you are currently controlling the velocity, could you repeat the experiments when controlling the acceleration? I expect to see more collisions.
>
> As described in Equation 4 of Appendix C, we do not control the velocity, which is constant instead for Dubins' car. The Dubins' car only controls a 1D action - the angular velocity of the heading angles. As shown in the video where a single agent controls the acceleration, the CAM agent learns to continuously decrease the velocity and stop in advance before a potential collision happens. Currently, we are also adding additional dynamic models of the vehicle in the larger environment under the multi-agent setting, and will add more results to evaluate the proposed methods.
>
>
>
> **Zip File:**
>
> /attachment/814e5bafa1a594ac3927e9564c8201ddea6daff2.zip

---

> > ### Comment · Reviewer_XUE1 · 2022-08-26
> > **additional comments**
> >
> > I appreciate the additional experiment, but I do think that additional data would be needed to understand the benefit of the method. I think that:
> > 1) it is concerning that in the single integrator example 9% of the points of the boundary do not satisfy the forward invariance condition. In this simple example, we could just pick the velocity vector from equation (6) in Appendix I to be zero to satisfy the invariance condition.
> > 2) For the Dynamic Dubins example, the learned invariant is not shown in the figure and no data is provided on how often the invariance condition is violated on the boundary.
> > 3) For the Dynamic Dubins example, the maximum velocity is 1 and the maximum deceleration is -1. Thus, the vehicle can be stopped in two time steps. I would like to see an example where the maximum velocity is 10 and maximum deceleration -1, as in this settings the controller should plan a few time steps ahead to avoid a collision. Also, it would be helpful to plot the velocity and input associated with the closed-loop trajectory.
> >
> > For the above reasons, I confirm my previous Recommendation.

---

> > > ### Author Response · Authors · 2022-08-27
> > > **additional data**
> > >
> > > **Comment:**
> > >
> > > Thank you very much for reading our response. We believe there are some misunderstandings in our additional results, and we wish to clarify them here. We also provide new results applying dynamic Dubins under the multi-agent navigation setting. The updated Appendix and two videos are attached to this response.
> > >
> > > > it is concerning that in the single integrator example 9% of the points of the boundary do not satisfy the forward invariance condition. In this simple example, we could just pick the velocity vector from equation (6) in Appendix I to be zero to satisfy the invariance condition.
> > >
> > > It’s 9‰, not 9%. We’ve updated it to 0.9% in the Appendix to avoid confusion. Again, we wish to emphasize that these states are out of the interest of our usage of CAM. We only focus on the forward-invariance property for those states on possible trajectories.
> > >
> > > > For the Dynamic Dubins example, the learned invariant is not shown in the figure and no data is provided on how often the invariance condition is violated on the boundary.
> > >
> > > We have updated Figure 9 to show the learned invariant. We have updated the text description. Around 98.6% of transitions along the trajectories are admissible and forward invariant. About 0.4% of states violate the forward-invariance property, i.e., the CAM agent crosses the boundary and enters the inadmissible region after executing the action. 1% of transitions are inadmissible, but all of them return to the admissible areas after executing sequences of most admissible actions.
> > >
> > > > For the Dynamic Dubins example, the maximum velocity is 1 and the maximum deceleration is -1. Thus, the vehicle can be stopped in two time steps. I would like to see an example where the maximum velocity is 10 and maximum deceleration -1, as in this settings the controller should plan a few time steps ahead to avoid a collision.
> > >
> > > Note that we first multiply the acceleration with $dt=0.05$ before adding it to the velocity (Line 651). As a result, the maximum velocity is 1, and the maximum deceleration is -0.05. As shown in the video, the CAM agent learns to continuously decrease the velocity and stop in advance before a potential collision happens.
> > >
> > > > Also, it would be helpful to plot the velocity and input associated with the closed-loop trajectory.
> > >
> > > We’ve updated the attached video to show the velocity and the control input along the trajectories in detail; please check.
> > >
> > >
> > > ### **Additional Results: Multi-Agent Navigation with Dynamic Dubins**
> > >
> > > Based on your previous suggestion, we have provided an additional result with dynamic Dubins’ car under the multi-agent navigation setting in the updated Appendix K. We also attach a new video showing the performances of our algorithm using this new dynamics model, with 3, 8, 32, 256, and 512 agents. Our observation in this new environment stays consistent with the main paper: Our CAM method outperforms other methods significantly in terms of the cumulative reward, while maintaining the highest safety rates of 99.5%-99.8%.
> > >
> > > We will be sure to update our paper with this new result. Thanks so much for your time.
> > >
> > >
> > > **Zip File:**
> > >
> > > /attachment/c2e2453d908d319f5a2a046c9375cfb67cd81f45.zip

---

### Official Review · Reviewer_759H · 2022-08-02

**Originality:** Very Good
**Technical Quality:** Very Good
**Clarity Of Presentation:** Very Good
**Impact:** 4

**Recommendation:**

Weak Accept: I recommend accepting the paper, but will not argue for my recommendation if the majority of other reviewers have a different opinion.

**Summary:**

This paper proposes learning a model of admissible controls that would avoid violating specified constraints in an MDP.  The paper calls these models Control Admissibility Models (CAMs), and demonstrate how these can be easily composed to provide a set of admissible actions at any state.  The CAMs are built using a Graph Neural Network, and trained using a time-based back-propoagation, where action that bring into a state that has no admissible actions, even if the state itself is admissible, are also included in the non-admissible state set.

Experiments are performed on a UR5 multi-arm task, a multi-agent driving task, a point-mass chasing task and a drone flight task.  Experiments show significant ability to maintain safety conditions as the number of agents increases, especially in the Car setting.

**Issues:**

I have mentioned above one blocking issue for me is the discussion of the use of reward, although I recommend an acceptance it is absolutely contingent on addressing this point.  I would also like a brief mention of inference time for the method, as this can drastically impact the types of systems on which it can run.

**Quality Of The Limitations Section:**

Limitations are addressed clearly

**Reviewer Expertise:**

3: The reviewer is fairly confident that the evaluation is correct

**Robotics Focus:**

Highly relevant to robotics but no hardware experiments

**Strengths And Weaknesses:**

This paper presents an interesting approach to dealing with dense multi-agent systems, and seems to have promissing results although I'm not an expert in this area and so don't have much experience in what is achievable.

One big weakness however is that I couldn't figure out how the paper uses reward from the environment.  It's clearly taken into account otherwise nothing would work, but the paper seems to skip over this important point or I somehow missed it after multiple re-reads, in which case this should definitely be fixed.

I would also appreciate a small mention of the inference time of this setup, to understand to what types of systems it could be deployed (i.e. up to what control frequency could it likely function).

**Summary Of Recommendation:**

I think this method is easily applicable to real systems, and provides a believable story on how to deal with a large number of interacting agents, which is an important problem in robotics.

---

> ### Author Response · Authors · 2022-08-23
> **Response to Reviewer 759H**
>
> Thank you for your careful reading. We are glad that you found our work interesting. We address your questions as follows.
>
> > how the paper uses reward from the environment
>
> In most navigation problems there are two types of requirements:  reaching goals (“reach” or goal-reaching requirements) and avoiding collision (“avoid” or safety requirements). These two types of requirements are in conflict with each other: to try reaching goal in the shortest path the robot may ignore obstacles and become unsafe, and to be safe the robot may simply stop moving and fail to reach its goals. Consequently, it is often hard to balance between these two types of rewards in a single reward function, and in multiagent problems in particular, naive designs typically lead to unsafe behaviors or deadlocks.
>
> We avoid reward engineering by decoupling the goal-reaching and collision-avoidance properties. For collision avoidance, we use CAMs to learn a set of safe actions at each state. For goal reaching, we rely on a given preference function that can simply be the Euclidean distance between the agent and the goal. The preference function can be simple because it will only pick out a preferred action from the set of safe actions. In this way, we decouple the two types of properties and avoid choosing weights between the two in a single scalar form in reward engineering.
>
> Consequently, our reward is designed as follows. The reward is non-zero only when the agent enters the danger zone (reward=-1) or goal region (reward=10). We use these rewards to label whether each transition is in the danger region (Line 10 in Algorithm 1).
>
> We will update the paper to claim the above statement more clearly.
>
> > I would also appreciate a small mention of the inference time of this setup, to understand to what types of systems it could be deployed (i.e. up to what control frequency could it likely function).
>
> We have provided the inference time for both Car and Drone in the attached PDF. As shown in the figure, the running time for each agent takes at most 4 milliseconds per step. We believe the running time can be further improved in real-world applications. Since our GNN does not require inter-agent communication, the agents can calculate the admissible scores independently and asynchronously with their own devices. Please check further discussions in the attached PDF.

---

### Official Review · Reviewer_JipF · 2022-08-02

**Originality:** Fair
**Technical Quality:** Fair
**Clarity Of Presentation:** Good
**Impact:** 3

**Recommendation:**

Weak Reject: I recommend rejecting the paper, but will not argue for my recommendation if the majority of other reviewers have a different opinion.

**Summary:**

In this paper, the authors study the multi-agent navigation problem and propose a control admissibility model (CAM) to output and compose the action instead of directly approximating the optimal policy. To this end, the navigation problem is represented by a graph, and graph neural network (GNN) is employed to compute the admissibility score. The CAM is validated on three navigation benchmarks.

**Issues:**

1. In the graph decomposition stage, the sub-graph will be different. How can a single GNN-based CAM deal with graphs with different vertices and edges?

2. In algorithm 1, it shows that CAM does not use reward to guide learning, which can also be seen from Eq. 2. How can CAM learn to navigate if no reward, i.e. goal information provided?

3. What is the influence of three key hyper-parameters in Eq. 2?

4. In Eq. 2, the authors claim the third term enforces the continuity of the CAM values on two consecutive transitions. How to understand it?

5. In the experiments, why the single agent algorithm DDPG outperforms the multi-agent algorithm baseline MACBF?

**Quality Of The Limitations Section:**

Limitations are addressed clearly

**Reviewer Expertise:**

3: The reviewer is fairly confident that the evaluation is correct

**Robotics Focus:**

Relevant but unlikely to deploy to hardware in near future

**Strengths And Weaknesses:**

This paper is well-written and easy to follow. The authors propose a simple but potentially effective framework, which breaks down the difficult overall optimal policy learning to multiple admissibility policies. However, the main contributions are not clear. First, the authors mainly introduce two multi-agent navigation difficulties, i.e. ad hoc reward engineering and poor generalization capacity. But why the GNN and decomposition-based CAM can deal with the above challenges is unclear. Second, the proposed method seems independent of the multi-agent system. In the multi-agent navigation problem, what are the key problems can CAM tackle? Additionally, the experiments are less convincing since only one multi-agent navigation baseline is compared.

**Summary Of Recommendation:**

Although the proposed method is clear and seems potentially effective, the key multi-agent navigation problem to be solved and the empirical results are not clear. Therefore, I suggest rejection of the current version.

---

> ### Author Response · Authors · 2022-08-23
> **Response to Reviewer JipF (1/2)**
>
> Thank you very much for your questions. We appreciate your constructive suggestions.
>
> > … First, the authors mainly introduce two multi-agent navigation difficulties, i.e. ad hoc reward engineering and poor generalization capacity. But why the GNN and decomposition-based CAM can deal with the above challenges is unclear.
>
> - Reward engineering: In most navigation problems there are two types of requirements:  reaching goals (“reach” or goal-reaching requirements) and avoiding collision (“avoid” or safety requirements). These two types of requirements are in conflict with each other: to try reaching goal in the shortest path the robot may ignore obstacles and become unsafe, and to be safe the robot may simply stop moving and fail to reach its goals. Consequently, it is often hard to balance between these two types of rewards in a single reward function, and in multiagent problems in particular, naive designs typically lead to unsafe behaviors or deadlocks.
>
>     We avoid reward engineering by decoupling the goal-reaching and collision-avoidance properties. For collision avoidance, we use CAMs to learn a set of safe actions at each state. For goal reaching, we rely on a given preference function that can simply be the Euclidean distance between the agent and the goal. The preference function can be simple because it will only pick out a preferred action from the set of safe actions. In this way, we decouple the two types of properties and avoid choosing weights between the two in a single scalar form in reward engineering.
>
> - Generalization for scalability. To generalize to a larger number of agents at inference time, we utilize the compositionality of the CAM and the graph representation. We decompose the test task into a group of subtasks that are seen during training and then compose the original admissible set using an intersection over all the admissible sets from these subtasks. Our main experiment results show that the proposed CAM generalizes notably well up to 512 agents, while the training task only has 3 agents.
>
> > In the multi-agent navigation problem, what are the key problems can CAM tackle?
>
> The key problem that CAM tackles is the generalization for scalability. Conventional learning-based multi-agent methods often fail for such generalization, when the distribution for the training tasks and inference tasks mismatch, e.g., training with 3 agents and inference with 512 agents.
>
> The generalization would be greatly improved if we could decompose the inference task into subtasks that are seen during training and then compose the admissible actions for all subtasks together. However, it is hard for traditional RL methods to compose the optimal actions gathered from these subtasks, since the composition operation is undefined. Apparently, we can take the average over these actions, but we doubt its effectiveness.
>
> CAM predicts a set of admissible actions for each subtasks. It is easy to compose these sets by taking the intersection. As described in Section 2.4, intuitively, the actions in this intersection should be admissible to all subtasks. We have shown the effectiveness of the decomposition-based CAM in the main experiment.
>
> > Additionally, the experiments are less convincing since only one multi-agent navigation baseline is compared.
>
> Our DDPG with GNN is a multi-agent algorithm. As mentioned in the main paper and Appendix F, we re-implement the DDPG with GNNs. The GNN actor takes an egocentric graph for each agent and outputs the corresponding action. The GNN critic takes the graph and action for each agent, and outputs the Q value. In such a case, the DDPG with GNN perceives the states of neighbor agents by taking the egocentric graph as the input, which makes it a multi-agent algorithm.
>
> Our GNN implementation deviates from the MA-DDPG algorithm (which is a standard multi-agent extension for DDPG) because of scalability concerns. In MA-DDPG, the actor network for each agent is independent and does not share the weight. As a result, MA-DDPG requires the same number of agents for the training and inference tasks. However, in our experiment, since we focus on the generalization for scalability, the number of agents during inference varies, while our training tasks are fixed to have three agents. Thus, it is impossible to deploy the MA-DDPG algorithm.
>
> We will update the description in both the main paper and the Appendix to reflect our GNN implementation of DDPG and MACBF better.

---

> ### Author Response · Authors · 2022-08-23
> **Response to Reviewer JipF (2/2)**
>
> > How can a single GNN-based CAM deal with graphs with different vertices and edges?
>
> The ability to take an arbitrary graph with different vertices and edges as the input is a general property of the GNN. Essentially, for each node in the graph, the GNN uses an aggregation function to aggregate the information from neighbor nodes, where such aggregation function can perform on a set with an arbitrary number of elements, e.g., sum / max / mean. For further details, please check how we design the GNN in Section 2.2.
>
> During training, the vertices and edges vary as well, since the neighbors within the observation radius often change when unrolling the trajectory. Please check Section 2.2 and Appendix B to D for more details about how we represent the graph.
>
> > How can CAM learn to navigate if no reward, i.e. goal information provided?
>
> If there is no goal information, the preference function can just be trivial (no preference means any/random safe action can be chosen), and the CAM will learn to navigate safely without goal-reaching behaviors.
>
> In multi-agent navigation, there are two types of properties to be satisfied: goal-reaching and collision-avoidance. In practice, previous works use rewards to balance these two properties, often requiring extensive reward engineering. We do not design such dense reward in this work, but focus on how to decouple these two properties. We design CAM for collision avoidance and use a simple preference function for goal-reaching. Using such decoupling, we can use CAM to navigate safely and effectively by choosing the most preferred admissible action at each time step.
>
>
> > What is the influence of three key hyper-parameters in Eq. 2?
>
> The three key hyper-parameters allow us to trade off the relative importance of each of the terms in the optimization. If the first two parameters are larger, the margin between the scores of admissible and inadmissible actions will be wider. If the third parameter is larger, then the CAM will be more conservative because most of the actions along the trajectory will remain a relatively high score.
>
> > In Eq. 2, the authors claim the third term enforces the continuity of the CAM values on two consecutive transitions. How to understand it?
>
> We apologize for this misleading sentence. Here we mean that once all pairs of consecutive transitions satisfy this condition, the transitions would form a forward invariant set, and any trajectory starting from inside the invariant set will never cross the admissible boundary. We will update the related text to explain the intuition better.
>
> > why the single agent algorithm DDPG outperforms the multi-agent algorithm baseline MACBF?
>
> As mentioned above, the DDPG we implement is a multi-agent algorithm. MACBF often fails because its objective forces it to imitate an expert goal-reaching controller. Such imitation might not be effective, when the collision avoidance property desires an action that deviates far from the goal-reaching controller.

---

### Meta-Review · Area_Chair_LQLH · 2022-08-14

**Recommendation:** Accept (Poster)
**Confidence:** 3

**Metareview:**

The reviewers appreciated the well-written paper and pointed out the ambitious overall objective. They found the proposed method to be an interesting and well-designed approach for dealing with dense multi-agent systems. However, they also raised a number of issues. To address these, the authors should try to improve the explanation of why their approach is actually suitable to tackle the major challenges associated with multi-agent navigation, i.e. why the method is better able to address ad-hoc reward engineering and poor generalization capacity. In addition, it should be clarified, how rewards from the environment are actually taken into account in the approach. To judge applicability, it would be beneficial to mention inference times of the algorithm. Finally, more experimental results would be needed to confirm that the method is working, and to judge how well the method approximates invariant sets. Please note the detailed lists of issues and suggestions of reviewers JipF and XUE1 in this context.

===

Post rebuttal/discussion update:

The authors provided substantial further input and clarifications, including additional experiments which addressed the main concerns of the reviewers. The presented method appears to be a useful and novel contribution for the community.

---

> ### Author Response · Authors · 2022-08-23
> **Response to Meta Review**
>
> We thank the meta-reviewer for the suggestions. We have responded to the reviewers in detail and added more information. In particular:
> - “why the method is better able to address ad-hoc reward engineering and poor generalization capacity”: We have explained our contributions in detail in the response to Reviewer JipF.
> - “how rewards from the environment are actually taken into account in the approach”: We have clarified how we avoid reward engineering in a decoupling way using the CAM and the preference function in the responses to Reviewer JipF and 759H.
> - “inference times”: Further details on the inference time have been provided in the attached Appendix in the response to Reviewer 759H.
> - “more experimental results”: We have provided several new experiment results based on reviewers’ suggestions. For further details, please check our response to Reviewer XUE1.
> - “detailed lists of issues and suggestions of reviewers JipF and XUE1”: For further discussions, please check the corresponding answers. We will improve the paper based on the suggestions.